Evaluation of transport conditions for autologous bone marrow-derived mesenchymal stromal cells for therapeutic application in horses

Espina Miguel 1
Jülke Henriette 2
Brehm Walter 1
Ribitsch Iris 2 3
Winter Karsten 2 4
Delling Uta 1 delling@vetmed.uni-leipzig.de
1 Large Animal Clinic for Surgery, Faculty of Veterinary Medicine, University of Leipzig , Leipzig , Germany
2 Translational Centre for Regenerative Medicine (TRM), University of Leipzig , Leipzig , Germany
3 Equine Clinic, University of Veterinary Medicine Vienna , Vienna , Austria
4 Institute of Anatomy, Faculty of Medicine, University of Leipzig , Leipzig , Germany
Esteban María Ángeles
Electronic publication date: 2016 Mar 22
Publication date: 2016
Volume: 4
Electronic Location ID: e1773
Received 2015 Nov 7; Accepted 2016 Feb 17
Copyright: ©2016 Espina et al.
Copyright year: 2016
Copyright holder: Espina et al.
License: This is an open access article distributed under the terms of the Creative Commons Attribution License, which permits unrestricted use, distribution, reproduction and adaptation in any medium and for any purpose provided that it is properly attributed. For attribution, the original author(s), title, publication source (PeerJ) and either DOI or URL of the article must be cited.
License URL: https://creativecommons.org/licenses/by/4.0/

Keywords: Horse, Mesenchymal stromal cells (MSCs), Transport, Viability

Funding: The authors received no funding for this work.

==============================
Background. Mesenchymal stromal cells (MSCs) are increasingly used for clinical applications in equine patients. For MSC isolation and expansion, a laboratory step is mandatory, after which the cells are sent back to the attending veterinarian. Preserving the biological properties of MSCs during this transport is paramount. The goal of the study was to compare transport-related parameters (transport container, media, temperature, time, cell concentration) that potentially influence characteristics of culture expanded equine MSCs.

Methods. The study was arranged in three parts comparing (I) five different transport containers (cryotube, two types of plastic syringes, glass syringe, CellSeal), (II) seven different transport media, four temperatures (4 °C vs. room temperature; −20 °C vs. −80 °C), four time frames (24 h vs. 48 h; 48 h vs. 72 h), and (III) three MSC concentrations (5 × 106, 10 × 106, 20 × 106 MSC/ml). Cell viability (Trypan Blue exclusion; percent and total number viable cell), proliferation and trilineage differentiation capacity were assessed for each test condition. Further, the recovered volume of the suspension was determined in part I. Each condition was evaluated using samples of six horses (n = 6) and differentiation protocols were performed in duplicates.

Results. In part I of the study, no significant differences in any of the parameters were found when comparing transport containers at room temperature. The glass syringe was selected for all subsequent evaluations (highest recoverable volume of cell suspension and cell viability). In part II, media, temperatures, or time frames had also no significant influence on cell viability, likely due to the large number of comparisons and small sample size. Highest cell viability was observed using autologous bone marrow supernatant as transport medium, and “transport” at 4 °C for 24 h (70.6% vs. control group 75.3%); this was not significant. Contrary, viability was unacceptably low (<40%) for all freezing protocols at −20 °C or −80 °C, particularly with bone marrow supernatant or plasma and DMSO. In part III, various cell concentrations also had no significant influence on any of the evaluated parameters. Chondrogenic differentiation showed a trend towards being decreased for all transport conditions, compared to control cells.

Discussion. In this study, transport conditions were not found to impact viability, proliferation or ability for trilineage differentiation of MSCs, most likely due to the small sample size and large number of comparisons. The unusual low viability after all freezing protocols is in contrast to previous equine studies. Potential causes are differences in the freezing, but also in thawing method. Also, the selected container (glass syringe) may have impacted viability. Future research may be warranted into the possibly negative effect of transport on chondrogenic differentiation.

Introduction

Mesenchymal stromal cells (MSCs) are increasingly applied for various diseases in humans (Sharma et al., 2014) and animals (Guercio et al., 2012; Smith, Garvican & Fortier, 2014). In the equine patient, MSCs are used locally, predominantly to treat tendon and ligament injuries (Godwin et al., 2012; Smith et al., 2013). The benefit of MSC injection to treat joint diseases is less well established (Ferris et al., 2014). Mesenchymal stromal cells can be isolated from multiple body tissues including bone marrow (BM), adipose tissue and blood (Sharma et al., 2014; Smith, Garvican & Fortier, 2014). Following isolation, MSCs need to be expanded in culture to provide adequate cell numbers for clinical use. Shipping conditions for transportation of expanded cells to the patient should ensure stem cell viability and maintenance of original cell characteristics, such as purity, identity, differentiation, and proliferation capacity. Various shipping conditions are being used for the transport of expanded MSCs to the equine patient, but details of validation experiments in horses are either not reported (Godwin et al., 2012), only unfrozen conditions were evaluated (Bronzini et al., 2012; Mercati et al., 2014) or only one frozen condition was evaluated (Garvican et al., 2014). Specifically, Bronzini et al. (2012) used blood-derived MSCs (n = 10) and compared 10 different transport media at 4 °C, room temperature (RT), and 37 °C for up to 72 h using “sterile tubes” of unspecified origin. Mercati et al. (2014) however, used fat-derived MSCs (n = 2), assessed one transport media at 4 °C vs. RT, for 24 h and 48 h; the origin of the transport container was not specified either. Finally, Garvican et al. (2014) evaluated BM-derived MSCs (n = 3) in 7 different media at 4–8 °C and one medium at −78 °C for up to 72 h, using a single type of specified cryotubes. A recent study focused specifically on short-term (2–5 d) cryopreservation (liquid nitrogen) of equine BM-derived MSCs (n = 9) prior to implantation (Mitchell et al., 2015). In this study six different transport media compositions were tested including different serum preparations, varying concentrations of dimethyl sulfoxide (DMSO), and culture medium. There were no significant differences between the different media. The authors concluded that clinicians may prefer a combination of autologous serum and low DMSO concentration for frozen MSC transport prior to clinical use.

When shipping unfrozen cells, it appears that RT is superior to 37 °C or 4 °C when shipping times do not exceed 12 h (Bronzini et al., 2012). With longer shipping times, keeping the cells at 4 °C resulted in higher viability compared to RT (Mercati et al., 2014). Further, superior results were obtained using phosphate buffered saline (PBS) compared to culture medium with or without blood serum or fetal bovine serum (FBS) as transport media at temperatures above 0 °C for up to 12 h (Bronzini et al., 2012). Others report no significant difference in cell viability after 12 h and 24 h in all transport media, but found the most rapid decline in viability over time in suspensions containing biological fluids such as BM aspirate, platelet-rich plasma or serum. Interestingly and in contrast to Bronzini et al. (2012), the least decline in viability was observed with culture medium containing FBS (Garvican et al., 2014). Transport of frozen equine MSCs (90% allogenic blood serum + 10% DMSO; −78 °C; up to 72 h) resulted in MSC viability of ∼80% (Garvican et al., 2014). A similar high viability (80–90%) was described by Mitchell et al. (2015). However, it should be emphasized that in both articles an invasive thawing method is described, which is unlikely to be practically feasible.

For the transport of frozen MSCs, it has to be considered that cryopreservation may impair stem cell functionality and engraftment of human blood-derived hematopoietic stem cells (Lioznov et al., 2008). Also, there have been reports of altered immunosuppressive properties of human MSCs following cryopreservation (François et al., 2012).

The important aspect of cell viability can be assessed in multiple ways, including manual or automated counting. Further, viability is usually expressed as percent of viable cells but the actual number of viable cells can be determined as well. Based on the four previous publications regarding transport of equine MSCs for immediate clinical use (Bronzini et al., 2012; Mercati et al., 2014; Garvican et al., 2014; Mitchell et al., 2015) no data of the total number of viable cells after a simulated transport are available.

Another unknown variable is the influence of the transport container, since plastic adherence is a key feature of MSC characteristics. Furthermore, it is known that both surface chemistry and biochemical signals affect MSC proliferation and differentiation (Almodóvar et al., 2010; Wang et al., 2015). Cryotubes are most frequently used in practical settings and in previous comparable studies (Garvican et al., 2014; Mitchell et al., 2015); others did not specify the container under investigation (Bronzini et al., 2012; Mercati et al., 2014).

Finally, the concentration of cells during transport may impact MSC characteristics. It has been stated that cell concentration may influence biological properties of hematopoietic stem cells (Dlimi, 2012). Also, cell concentration appears to have an impact on cell viability in non-MSC cell lines (De Loecker et al., 1998; Costa et al., 2000).

Therefore, the aim of this present study was to determine the impact of temperature during transport, transport media, transport times, transport containers, and MSC concentrations on equine, BM-derived MSC characteristics.

Our hypotheses were that (1) transport container, (2) transport media, temperature and time, and (3) MSC concentration have an effect on MSC viability, proliferation and trilineage differentiation capacity.

Material and Methods

Study design and general procedures

Study design

The study entailed three consecutive parts in which preceding results were applied in the subsequent steps (Fig. 1). In all conditions and controls, cell quality was assessed with respect to viability (percent, total number viable cells), proliferation and trilineage differentiation capacity. In part I, recoverable cell suspension volume for the different containers was additionally determined. The most suitable container was subsequently used throughout the study. In part II, different transport media were tested in positive (four types of transport media) and negative (three types of transport media) temperatures, each for two time frames (24 and 48 h, 48 and 72 h, respectively). This amounted to 28 different test conditions: 16 at temperatures above 0 °C and 12 within the negative temperature range. Finally, in part III, three different MSC concentrations were compared using the best transport media/temperature/time combination as determined in part II. A total of six test conditions were compared; three for the positive and negative temperature, respectively. Controls for each condition were the cell characteristics at time point 0, immediately before each test condition was started. In each of the three parts of the study, the conditions were always evaluated using MSCs of six different horses (n = 6). Further, all MSC differentiation protocols were performed in duplicates.

Figure 1 Study design of the three parts of the study.

Sampling of BM and venous blood

All experiments were approved by the State Animal Care Committee (V12/09, Landesdirektion Leipzig, Free State of Saxony, Germany). Bone marrow was harvested from the sternum of 11 Warmblood geldings. Seven horses were sampled twice and used for two parts of the study (over six months between the aspirations). The horses were aged between 15–16 years and the procedure was performed as previously described (Delling et al., 2012). Briefly, after intravenous sedation with 0.06 mg/kg romifidine (Sedivet; Boehringer Ingelheim, Ingelheim, Germany) and 0.02 mg/kg butorphanol (Torbugesic; Fort Dodge Veterinär, Würselen, Germany), the area of the sternum was aseptically prepared and locally anesthetized. A BM aspiration needle (Bone Marrow Harvest Needle; Angiotech, Gainesville, FL, USA) was advanced into the bone and 2 × 20 ml marrow was aspirated into heparinized syringes (10,000 IU heparin-sodium (B. Braun, Melsungen, Germany)/20 ml syringe). Further, 2 × 20 ml venous blood samples were obtained from the jugular vein for the production of autologous plasma using heparinized syringes as described above.

Processing of BM and venous blood

Within 4 h after sampling, the BM was centrifuged in 50 ml conical tubes (Falcon Tube; BD Biosciences, Erembodegem, Belgium; 10 min, 600 × g, 10 °C). The bone marrow supernatant (BMS) was collected, filtered (BD Falcon cell strainer, BD Bioscience), and stored at −80 °C for further use. From the remaining cellular fraction of the BM, mononuclear cells (MNCs) were isolated by density gradient centrifugation. Briefly, the cellular fraction was diluted with PBS (PAA, Pasching, Austria), layered on 15 ml Ficoll-Paque Premium (GE Healthcare, Uppsala, Sweden) and centrifuged (20 min, 1,000 × g, 20 °C). The buffy coat containing MNCs was collected, washed twice with PBS and seeded in one T75 tissue culture flasks (Greiner Bio-One, Frickenhausen, Germany) containing culture medium, i.e., Dulbecco’s modified Eagle Medium (DMEM) low-glucose (PAA) supplemented with 10% FBS (Sigma Aldrich, Steinheim, Germany), 1% penicillin/streptomycin (PAA) and 8.9 µg/ml ascorbic acid (Sigma). Cell cultures were kept at 37 °C in a humidified atmosphere at 5% CO2. After 24 h, the adherent cells were washed with PBS and culture medium was added again. For the remaining expansion period, culture medium was changed twice a week until cells reached subconfluency (p0).

In parts I and III of the study, cryopreserved MSCs were used, whereas fresh cells were evaluated in part II. For the latter, harvested MSCs of p0 were further expanded until p3 by plating 500,000 MSC/T175 tissue culture flasks and maintaining the above mentioned culture conditions. In contrast, MSCs for parts I and III were cryopreserved at p0 for later use. For this, MSC (p0) suspension was centrifuged (5 min, 600 ×g, 4 °C) and the obtained cell pellet was resuspended in FBS supplemented with 10% DMSO (Sigma Aldrich, Steinheim, Germany). The suspension was transferred into cryotubes (Greiner Bio-One, Frickenhausen, Germany) and temperature was decreased stepwise at a rate of 1 °C/min to −80 °C using a freezing container (Nalgene Mr. Frosty; Nalge Nunc, Rochester, NY, USA). After 12 h maintenance at −80 °C in a freezer, the temperature was lowered further to −150 °C in a freezer where cells were stored until further use. For thawing, the samples were placed in a 37 °C water bath. The MSCs were transferred to conical tubes, washed with PBS and centrifuged (5 min, 600 ×g, 4 °C). The obtained cell pellet was suspended in culture medium and expanded as mentioned above until p3.

A hemocytometer (Neubauer-improved, Marienfeld-Superior, Lauda–Königshofen, Germany) was used for cell counting throughout the study.

Autologous blood plasma was harvested and stored in an identical manner as the BMS.

MSC viability testing

Cell viability was determined using Trypan Blue stain (Sigma) and the fraction of unstained (viable) cells was determined manually. Viability was expressed as percent of viable cells; additionally, the total viable cell count per condition was determined.

Proliferation capacity

To allow comparison of MSC growth characteristics, cumulative doubling rates were calculated for each condition as previously described (Giovannini et al., 2008). For this, cells (p4) were plated at a density of 500 cells/cm2 on T25 tissue culture flasks in culture medium. After seven days in culture, the total number of MSCs per sample was assessed. The process was repeated at weekly intervals until p7. The population doublings were calculated with the equation PD = log10 (N∕N0) ×3.33, where N is the number of harvested cells and N0 the number of plated cells. Cumulative population doublings (cPD) for each time point were calculated by adding values of all previous PDs.

Figure 2 All parts of the study: representative histological images of the scoring system to assess adipogenic, early and late osteogenic (scale bar 100 µm), and chondrogenic differentiation (scale bar 200 µm).

Trilineage differentiation potential

For adipogenesis, a modified protocol was used (Giovannini et al., 2008). Briefly, cells (p4) were seeded in duplicate at a density of 5,000 cells/cm2 in 12-well plates (Greiner). At subconfluency, induction medium was added consisting of DMEM/Ham’s F12 (PAA), 5% rabbit serum (Sigma), 10% FBS, 1% penicillin/streptomycin, 1 µM dexamethasone (Sigma), 100 µM indomethacine (Sigma) and 500 µM 3-IBMX (3-isobutyl-1-methylxanthine; Sigma), as well as 1,700 nM insulin (Invitrogen, Karlsruhe, Germany) and maintained for 72 h. Afterwards, cells were fixed with 4% paraformaldehyde (Roth, Karlsruhe, Germany) and the production of intracellular lipid droplets was assessed semi-quantitatively by 0.35% Oil red O staining (Sigma) counterstained with Mayer hematoxilin (Sigma) and assigning scores of 0–3 for no, low, medium or high numbers of visible fat droplets. Representative images are depicted in Fig. 2.

For osteogenesis, cells (p4) were seeded in duplicate at a density of 5,000 cells/cm2 in 12-well plates. At subconfluency, differentiation was induced in DMEM/Ham’s F12, 10% FBS, 1% penicillin/streptomycin supplemented with 10 nM dexamethasone, 10 mM ß-glycerophosphate (Sigma) and 0.1 mM ascorbic acid. The medium was changed twice a week. Early and late osteogenesis was evaluated using an alkaline phosphatase stain (Sigma) and Alizarin red stain (Sigma) at day 14 and 21 after induction, respectively. Both stains were assessed semi-quantitatively by assigning scores of 0–3 for no, low, medium and high blue intracellular staining and red stained-extracellular calcium deposition, respectively. Representative images are depicted in Fig. 2.

Chondrogenesis was induced in micromass pellet cultures (p4) (Giovannini et al., 2008). To exclude non-viable cells after the simulated transport, all recovered cells were cultured routinely for 12 h. Following, only adherent cells were harvested and pellets were prepared as 5 × 105 cells in duplicate, placed in 15 ml tubes and centrifuged (5 min, 280 × g, 4 °C). Pellets were incubated in chondrogenic medium consisting of DMEM high glucose (4.5 g/l; PAA) supplemented with 1% ITS + Premix (BD Biosciences), 0.1 µM ascorbic acid, 0.4 µM proline (Roth), 100 nM dexamethasone and 10 ng/ml human recombinant TGF-ß1 (BD Biosciences). Medium was replaced twice a week. After 21 days in chondrogenic culture, pellets were fixed for 24 h in 4% paraformaldehyde, dehydrated, embedded in paraffin and cut into 4 µm thick sections. Histologic sections were stained with Safranin O-fast green (Sigma) and Alcian blue (Sigma) to detect extracellular glycosaminoglycan depositions. Masson trichrome staining (Sigma) was performed to evaluate collagen synthesis. All slices were assessed semi-quantitatively using an established scoring system (Bern Score) based on Safranin O-fast green stains and with a maximal score of nine (Grogan et al., 2006). A staining scale from 0 to 3 was used for Alcian blue and Masson trichrome stained sections, respectively. Representative images are depicted in Fig. 2.

All histologic slides were scored at 10–20×magnification. Evaluation of the chondrogenic differentiation was blinded to experimental conditions, whereas complete blinding of the adipogenic and osteogenic differentiation was impeded by the consecutive order in the 12-well plates known to the investigator.

Part I: Evaluation of transport containers

Each condition was tested using BM samples of six horses. Cells of p3 were divided into six aliquots. The first aliquot served as control (time point 0) without further manipulation. The remaining five aliquots (“transport groups”) were transferred to five different transport containers: (1) cryotubes (Art No 126263; Greiner), (2) plastic syringe/plastic tipped plunger (Injekt Innohep, B. Braun), (3) plastic syringe/rubber tipped plunger (Art No 370-003, Henry Schein Vet, Hamburg, Germany), (4) glass syringe/rubber tipped plunger (Art No 663570002, 513010007, and 423040001; Gerresheimer, Düsseldorf, Germany), and (5) CellSeal (CellSeal, 2 ml Needleless Port Foil Covered- Long Tubing Art No 10223-700-47; General Biotechnology, Indianapolis, IN, USA) (Fig. 3). The standardized conditions for the “transport groups” entailed a cell concentration of 10 × 106 MSC/ml, use of 500 µl cell suspension per container (5 × 106 MSC/container), DMEM high glucose + 20% FBS as transport media, and the maintenance for 24 h at RT (21 °C) under light protection within the laboratory. After 24 h, containers were gently agitated, the contents expelled from the containers into conical tubes and the recoverable volume was measured using a micropipette. Viability, proliferation and differentiation capacity were assessed as described above. Subjective handling characteristics of the individual containers concerning content recovery and ease of use were recorded.

Figure 3 Part I: transport containers: (1) cryotube, (2) plastic syringe/plastic tipped plunger, (3) plastic syringe/rubber tipped plunger, (4) glass syringe/rubber tipped plunger, and (5) CellSeal.

Part II: Evaluation of transport media, temperature and time

The glass syringe was selected to be used as transport container in part II and III based on results of part I. Again, each condition was tested using BM samples of six horses. Cells of p3 were divided into 29 aliquots: one aliquot served as control (without additional manipulation) and 28 aliquots were used for further testing. Four different transport media, specifically (a) PBS, (b) autologous BMS, (c) autologous blood plasma, and (d) HypoThermosol FRS (HypoThermosol FRS; BioLife Solutions, Bothell, WA, USA), were tested at RT and 4 °C, each for a 24 h and a 48 h “transport” time. The temperature of 4 °C was maintained in a refrigerator. Three different transport media, specifically (a) CryoStor CS10 (BioLife Solution; which contains 10% DMSO), (b) autologous BMS + 10% DMSO, and (c) autolougous blood plasma + 10% DMSO, were tested at −20 °C and −80 °C, each for a 48 h and 72 h “transport” time. Again, each glass syringe contained 500 µl cell suspension at a concentration of 10 × 106 MSC/ml (5 × 106 MSC/container).

The freezing protocol was adapted for the glass syringe. Specifically, the MSC suspension was placed into the glass syringe via the syringe tip using a pipette. The tip was closed with a sterile cap (Art No 520010006; Gerresheimer). The plastic plunger was unscrewed from its rubber tip and the syringe covered loosely with flexible wrap (Parafilm; Bemis, Neenah, WI, USA). The ensuing freezing protocol is described above; the glass syringe without plunger fits into the freezing container similar to the cryotube. Temperatures of −20 °C and −80 °C were maintained in freezers with integrated thermometers. Thawing was also similar as described previously. Specifically, the glass syringe was placed in a 37 °C water bath until no ice was visible within the suspension. The flexible wrap was removed, the plunger screwed back on the rubber plunger tip and the content expelled for further evaluation. At the end of each simulated transport viability, proliferation and differentiation capacity were assessed as described above.

Part III: Evaluation of cell concentration

Conditions with the highest median cell viability in part II at temperatures above and below 0 °C, respectively (#5: BMS, 4 °C, 24 h; #26: CryoStor, −20 °C, 72 h) were further investigated in part III. Specifically, cells of p3 were divided and placed at concentrations of 5 × 106, 10 × 106, or 20 × 106 MSCs per ml transport media into glass syringes (1 ml suspension/container). Each condition was tested using BM samples of six horses. In contrast to part I and II, the control condition (time point 0) in this group additionally entailed the application of the MSC suspension into the glass syringe and immediate withdrawal from the syringe. This means that the control underwent the handlings of the “transport” itself, except for the actual time lapse of the transport. This step was added to specifically assess the influence of the manipulation using the syringe vs. the influence of the transport itself. The freezing and thawing of the MSC suspension within the glass syringe was performed identically as described in part II. Viability, proliferation and differentiation capacity were assessed as described above after 24 h (for temperatures above 0 °C) or 72 h (for temperatures below 0 °C).

Data analysis

Data were tested for normal distribution using the Shapiro–Wilk test. Comparisons between the individual conditions were made using Friedman- and Wilcoxon-Tests. Significance was set at p < 0.05. Post-hoc Bonferroni correction was applied to adjust the p-value. Data were expressed as median and interquartile range (IQR). Statistical analysis was performed with IBM SPSS Statistics (version 22, Armonk, NY, USA).

Results

Part I: Evaluation of transport containers

The recovered volume after “transportation” was highest from the glass syringes (container 4; 91%, 453 µl IQR 16 µl) and lowest from CellSeal (78%, 390 µl IQR 35 µl). Plastic syringes (2) and (3) yielded volumes of 88% (440 µl IQR 35 µl) and 90% (450 µl IQR 19 µl), respectively. The cryotubes yielded a volume of 83% (415 µl IQR 40 µl). Differences between the five containers were not significant. Figure 4 depicts the five containers after 24 h of “transport” and immediately before evacuation. Note the cell pellet which formed at the bottom of all containers. Subjectively, we noticed foam formation after the content had been agitated in the cryotube (container 1) and CellSeal (container 5), thus decreasing the recoverable volume. Upon pushing the plunger of the syringes to expel the cell suspension, the glass syringe allowed for smooth, controlled delivery of the cell suspension, whereas both plastic syringes tended to have an irregular resistance during content delivery.

Figure 4 Part I: representative images of the five transport containers after 24 h “transportation” and before gentle agitation to dissolve the cell pellet formed at the bottom of the container: (1) cryotube, (2) plastic syringe/plastic tipped plunger, (3) plastic syringe/rubber tipped plunger, (4) glass syringe/rubber tipped plunger, (5) CellSeal.

Viability assessed by Trypan Blue staining was lower in all transport groups compared to the control group, i.e., viability at time point 0 (76.8% IQR 14.6%), although this was not statistically significant. There was also no significant difference in viability (%) and the total number viable cells between the five containers (Table 1). The highest viability was observed in glass syringes, followed by the plastic syringe/rubber tipped plunger, the cryotube, CellSeal, and the plastic syringe/plastic tipped plunger.

Table 1 Part I: MSC viability of 5 × 106 MSCs after “transport” in five different containers expressed in percent (%) viable cells and total number viable MSCs recovered; no significant differences.

	Viability in % (median (IQR))	Total number viable MSC (median (IQR))	
Control (time point 0)	76.8 (14.6)	not applicable	
1-cryotube	38.1 (31.2)	1.03 × 106 (1.77 × 106)	
2-plastic syringe/plastic tipped plunger	37.4 (23.9)	1.85 × 106 (1.11 × 106)	
3-plastic syringe/rubber tipped plunger	38.9 (22.9)	1.97 × 106 (1.61 × 106)	
4-glass syringe/rubber tipped plunger	44.0 (20.3)	1.88 × 106 (1.13 × 106)	
5-CellSeal	38.0 (20.0)	1.43 × 106 (1.20 × 106)	

Proliferation capacity, adipogenic, early and late osteogenic differentiation as well as chondrogenic differentiation were not significantly affected by the containers and there were no differences compared to the control group. Results of the chondrogenic differentiation are given in Table 2.

Table 2 Part I: chondrogenic differentiation capacity of MSCs after “transport” in five different transport containers, expressed as median (IQR); container (1) cryotube, (2) plastic syringe/plastic tipped plunger, (3) plastic syringe/rubber tipped plunger, (4) glass syringe/rubber tipped plunger, and (5) CellSeal; no significant differences.

	Control	Container 1	Container 2	Container 3	Container 4	Container 5	
Bern score (Safranin-O; 0–9)	2.3 (0.9)	1.3 (4.1)	1.0 (1.1)	1.5 (3.1)	2.3 (2.5)	3.0 (2.8)	
Alcian Blue (0–3)	1.8 (0.5)	1.0 (0.8)	1.8 (0.9)	2.0 (0.8)	1.5 (0.4)	1.8 (0.5)	
Masson Trichrome (0–3)	2.0 (0)	1.5 (1.0)	1.5 (0.5)	2.0 (0.5)	1.8 (0.5)	1.5 (1.4)	

Based on the highest volume of recovery, subjective ease of use, and highest cell viability among the transport groups, the glass syringe (container 4) was chosen as being the most suitable container for the subsequent evaluation.

Part II: Evaluation of transport media, temperature and time

Viability in all test conditions was decreased compared to the control group (Fig. 5 and Table 3). Subjectively, at temperatures above 0 °C, cells generally had higher viabilities after 24 h of “transport” compared to 48 h of “transport.” Furthermore, cells kept at 4 °C tended to have higher viability than cells kept at RT. With respect to the transport media, we observed highest viabilities with BMS and plasma. None of these observations reached significance due to Bonferroni correction (significance level p < 0.002) and the large number of comparisons. The highest viability of 70.6% within the positive range was observed with condition #5 (BMS, 4 °C, 2 h). Regarding the total number viable MSCs higher cell counts were also obtained using BMS or blood plasma in suspension (Fig. 6). Specifically, the two highest numbers were 3.61 and 3.40 × 106 MSC (#7 BMS, RT, 24 h and #5 BMS, 4 °C, 24 h, respectively). Almost for all conditions the cell count decreased from 24 h to 48 h.

Figure 5 Part II: MSC viability.

(% Trypan Blue exclusion, median, IQR [box], range [whisker], ° outlier up to 1.5 × box lengths, * extreme values, outside 3 × box lengths) before (#0) and after various “transport” conditions in a positive temperature range (#1–16) and negative temperature range (#17–28); no significant differences (p < 0.002).

Table 3 Part II: MSC viability after various “transport” conditions in a positive temperature range and negative temperature range.

Viability is expressed in percent (%) viable cells and the total number viable MSCs recovered; no significant differences.

Rank according to % viability	Transport condition	Viability in % (median (IQR))	Total number viable MSCs (median (IQR))	
		Control	RT	0	75.3 (11.0)	Not applicable	
1	#5	BMS	4 °C	24 h	70.6 (8.4)	3.40 × 106 (1.17 × 106)	
2	#7	BMS	RT	24 h	67.5 (4.7)	3.61 × 106 (1.70 × 106)	
3	#9	Blood plasma	4 °C	24 h	63.0 (23.4)	2.73 × 106 (0.19 × 106)	
4	#11	Blood plasma	RT	24 h	55.2 (18.7)	2.66 × 106 (1.28 × 106)	
5	#6	BMS	4 °C	48 h	49.0 (6.3)	2.54 × 106 (1.48 × 106)	
6	#1	PBS	4 °C	24 h	46.7 (9.5)	2.46 × 106 (0.72 × 106)	
7	#10	Blood plasma	4 °C	48 h	45.0 (8.0)	2.93 × 106 (0.63 × 106)	
8	#15	HypoThermosol	RT	24 h	40.6 (14.7)	1.85 × 106 (1.03 × 106)	
9	#14	HypoThermosol	4 °C	48 h	39.2 (6.1)	2.36 × 106 (0.62 × 106)	
10	#13	HypoThermosol	4 °C	24 h	38.2 (11.1)	2.03 × 106 (0.96 × 106)	
11	#26	CryoStor	−20 °C	72 h	35.5 (16.4)	1.89 × 106 (1.13 × 106)	
12	#25	CryoStor	−20 °C	48 h	35.3 (3.6)	2.00 × 106 (0.78 × 106)	
13	#28	CryoStor	−80 °C	72 h	34.2 (11.8)	1.41 × 106 (1.16 × 106)	
14	#27	CryoStor	−80 °C	48 h	30.2 (13.8)	1.64 × 106 (1.26 × 106)	
15	#8	BMS	RT	48 h	28.6 (13.2)	1.53 × 106 (0.48 × 106)	
16	#2	PBS	4 °C	48 h	24.9 (5.6)	1.61 × 106 (0.53 × 106)	
17	#16	HypoThermosol	RT	48 h	24.8 (19.9)	1.00 × 106 (1.03 × 106)	
18	#3	PBS	RT	24 h	23.5 (32.1)	1.65 × 106 (1.56 × 106)	
19	#12	Blood plasma	RT	48 h	22.4 (15.4)	1.51 × 106 (0.81 × 106)	
20	#4	PBS	RT	48 h	20.4 (13.5)	0.85 × 106 (1.04 × 106)	
21	#21	Blood plasma	−20 °C	48 h	6.0 (0.7)	0.30 × 106 (0.14 × 106)	
22	#17	BMS	−20 °C	48 h	5.8 (5.7)	0.33 × 106 (0.11 × 106)	
23	#18	BMS	−20 °C	72 h	4.3 (1.7)	0.21 × 106 (0.06 × 106)	
24	#24	Blood plasma	−80 °C	72 h	4.1 (8.5)	0.14 × 106 (0.65 × 106)	
25	#22	Blood plasma	−20 °C	72 h	3.3 (5.5)	0.15 × 106 (0.35 × 106)	
26	#19	BMS	−80 °C	48 h	2.9 (3.8)	0.14 × 106 (0.14 × 106)	
27	#23	Blood plasma	−80 °C	48 h	2.0 (8.0)	0.09 × 106 (0.39 × 106)	

Within temperatures below 0 °C, the highest cell viability and the highest total number viable cells were found using CryoStor (#25–28). Among these, test condition #26 (CryoStor, −20 °C, 72 h) performed best, albeit not statistically significant. Proliferation capacity, adipogenic, early and late osteogenic differentiation were not affected by any of the “transport” conditions compared to the control group. However, data for conditions #17–24 was not collected due to insufficient cell numbers harvested after the simulated transport.

All tested conditions had lower Bern scores (Safranin O/Fast green stain) compared to the control group, indicating a decreased presence of mature proteoglycans (Fig. 7), but this was not statistically significant. Scores for the Alcian blue stain were similar to those for the Bern scores. Collagen deposition, evaluated on Masson-Trichrom-stains, was not affected by any of the tested conditions.

Figure 6 Part II: total numbers viable MSCs recovered.

(Trypan Blue exclusion, median, IQR (box), range (whisker), ° outlier up to 1.5 × box lengths, * extreme values, outside 3 × box lengths) after various “transport” conditions in a positive temperature range (#1–16) and negative temperature range (#17–28); the dotted line at 5 × 106 illustrates the cell count placed into the containers at time 0.

Figure 7 Part II: chondrogenic differentiation capacity.

Bern score indicating proteoglycan deposition (median, IQR (box), range (whisker)); a decrease in all test conditions compared to the control group (#0) was noted; no significant differences (p < 0.003); data for #16–24 are missing due to insufficient cell numbers after the “transport.”

Figure 8 Part III: MSC viability (% Trypan Blue exclusion, median, IQR (box), range (whisker), °outlier) under the influence of different cell concentrations before (control) and after “transport”.

Table 4 Part III: MSC viability after simulated transport using three different cell concentrations.

Viability is expressed in percent (%) viable cells and the total number viable MSCs recovered; in each condition 1 ml were used; no significant differences.

	Viability in % [median (IQR)]	Total number viable MSCs [median (IQR)]	
Control; RT; 0 h; 5 × 106 MSC/ml	82.6 (2.9)	2.84 × 106 (1.28 × 106)	
Control; RT; 0 h; 10 × 106 MSC/ml	83.6 (4.4)	6.98 × 106 (0.78 × 106)	
Control; RT; 0 h; 20 × 106 MSC/ml	81.0 (4.8)	14.45 × 106 (1.46 × 106)	
Bone marrow supernatant; 4 °C; 24 h; 5 × 106 MSC/ml	62.0 (8.8)	2.43 × 106 (0.25 × 106)	
Bone marrow supernatant, 4 °C, 24 h; 10 × 106 MSC/ml	62.1 (7.2)	5.38 × 106 (0.10 × 106)	
Bone marrow supernatant; 4 °C; 24 h; 20 × 106 MSC/ml	63.2 (7.3)	11.33 × 106 (1.11 × 106)	
CryoStor; −20 °C; 72 h; 5 × 106 MSC/ml	61.8 (9.7)	1.69 × 106 (0.23 × 106)	
CryoStor; −20 °C; 72 h; 10 × 106 MSC/ml	63.9 (6.2)	4.28 × 106 (0.49 × 106)	
CryoStor; −20 °C; 72 h; 20 × 106 MSC/ml	63.2 (7.0)	10.01 × 106 (2.33 × 106)	

Part III: Evaluation of cell concentration

We did not observe significant differences in viability compared to the control group or between any of the three evaluated MSC concentrations after transport, using conditions #5 (BMS, 4 °C, 24 h) and #26 (CryoStor, −20 °C, 72 h) (Fig. 8, Table 4), although subjectively, viability (%) following both transport conditions was consistently lower than in the control group. Importantly, in this part of the study, the control group MSCs was handled the same way as the “transport” group, but without the time lapse for the “transport.” It is remarkable, that manipulation alone, including the preparation for the cell counting procedure, resulted in a considerable loss of cells. The cell deficits in the control group were 27.8% (for 20 × 106 MSC/ml), 30.2% (for 10 × 106 MSC/ml) and 43.2% (for 5 × 106 MSC/ml).

Cell proliferation and trilineage differentiation capacity were not affected by the test conditions.

Discussion

The current study shows that transport of expanded MSCs can decrease cell viability to levels that would be unacceptable for clinical use. The aim of this study was to determine influence of different transport factors on MSC’s viability and ability to differentiate into different cell lineages. We were unable to identify significant differences between conditions tested, possibly due to the small sample number (n = 6) for each evaluated condition combined with the large number of comparisons between conditions that were performed in this study. Nevertheless, there are several aspects worth discussing.

An effect of transport media on cell survival within the positive temperature range has been reported before (Garvican et al., 2014). In this previous report equine MSC viability declined most rapidly in all allogenic biological fluids compared to culture medium and saline (Garvican et al., 2014). This is an interesting finding, because even though we did not reach significant results in our study, we observed contrary results: viability was better preserved with autologous BMS or plasma vs. PBS or HyoThermosol. A potential explanation might be complement system activation in allogenic but not autologous combinations resulting in increased MSC lysis. If autologous products are not available, thermal treatment (56 °C, 30 min) of allogenic BMS or blood plasma may resolve this problem. Interestingly, Garvican et al. (2014), who used allogenic products in the above mentioned study stated in their discussion, that pilot studies suggested “that heat treatment of allogeneic plasma, bone marrow aspirate and platelet-rich plasma (56 °C for 1 h) resulted in a small, but measurable improvement in cell viability (J. Dudhia, unpublished observation).” However, heat inactivation in turn adversely affects biological products as well (Giard, 1987) and might therefore be controversial. No difference was found in a study assessing equine MSC expansion in allogenic plasma lysate vs. FBS, i.e., a xenogenic medium (Seo et al., 2013). Unfortunately, no comments were made on complement inactivation in either transport media. In contrast to that, no influence on equine MSC survival was found when comparing ten different transport media encompassing PBS, DMEM, with or without the addition of 20% or 80% equine serum (presumably allogenic), or 20% or 80% FBS (Bronzini et al., 2012). Subsequently, PBS was recommended by the authors without further explanation. We used culture medium (DMEM high glucose + 20% FBS) in part I of our study. The observed low viability (37–44%) as well as the inclusion of FBS prompted us to investigate alternative transport media in part II. The addition of FBS to culture medium during transport of equine MSCs was omitted on purpose by others to avoid introducing foreign materials (Mercati et al., 2014). FBS is considered to potentially cause xenogenic immune reactions and may additionally carry the risk of transmitting bovine pathogens such as viruses, bacteria, and prions (Sundin et al., 2007). A lower degree of inflammation was observed after intraarticular injection of autologous MSCs compared to allogenic or xenogenic MSCs, thus reinforcing the importance of potential immunoreaction due to foreign proteins (Pigott et al., 2013).

We observed an unacceptably low viability after transporting MSCs within the negative temperature range particularly using BMS + 10% DMSO or blood plasma + 10% DMSO. This is in contradiction to previous studies where superior viability was obtained by freezing MSCs in 90% allogenic blood serum + 10% DMSO, i.e., viabilities between 60–80% after 48–72 h were observed (Garvican et al., 2014). Notably, after freezing equine MSCs in transport media containing either autologous or allogenic serum or FBS no significant differences in viability and growth characteristics were found (Mitchell et al., 2015). An overall high viability of 80%–90% post thawing was observed. Differences between protocols may explain contradictory findings to our study, for example the method of thawing. We submerged samples quickly in a 37 °C warm water bath, which is a realistic procedure for equine clinicians. Rapid thawing of mammalian cells is recommended to prevent ice crystal formation and cell lysis, thus, 37 °C is the recommended thawing temperature (Katayama et al., 1997; Phelan, 2007). Contrary, Mitchell et al. (2015) used a modified thawing method as a result of own pilot studies. Specifically, during the thawing process PBS was added to the MSC suspension to prevent post-thaw osmotic shock. They report viabilities less than <60% during their pilot studies when using a standard thawing protocol. Additionally, adding up to 20 ml PBS causes a dilution of the transport medium including the cytotoxic DMSO before the viability assessment was started; this is also in contrast to our study. A similar thawing method was used by Garvican et al. (2014) as well and might explain the superior viability following the freezing protocol. All of these invasive measures are no feasible options in equine clinical settings. It is very likely, that thawing is performed in an even less controlled manner using warm hands or pockets of pants.

In our study the choice of transport container may also have impacted viability during the freezing process. In part I, the containers were evaluated at RT only and as result a glass syringe with a rubber tipped plunger was used to freeze the MSC suspension. Previously published cryoprotocols for equine MSCs stated the use of cryovials or cryotubes. In our study the material (glass, rubber) or the handling within the syringe may have negatively influenced cell viability during the freezing or thawing process.

Another point to discuss is the use of DMSO as cryopreserving agent. DMSO was part of all transport media within the negative temperature range in our study; however, at concentrations used for cryopreservation and temperatures >4 °C DMSO is potentially cytotoxic. Adverse and toxic reactions in recipient human patients have been reported (Thirumala, Goebel & Woods, 2013). Finally, DMSO has been described as being capable of inducing differentiation of stem cells into cardiac or neuronal-like cells (Woodbury et al., 2000; Young et al., 2004). Even though Mitchell et al. (2015) did not identify a difference between MSCs after freezing in media containing 5% or 10% DMSO, no comparison was made to native MSCs before the DMSO contact.

Based on our data, no conclusion regarding the optimal transport media of choice for shipping frozen equine MSCs can be made due to the overall low viability observed. The use of CryoStor merits caution since to our knowledge no data are available on in vivo compatibility of this product after, for example, intratendinous or intraarticular injection in horses. An alternative concept practiced after transportation of human hematopoietic stem cells, is washing the MSCs upon arrival, but this procedure is not feasible for most practicing veterinarians.

The subjectively observed declining viability of unfrozen MSCs over time (24 h vs. 48 h), which was seen in our study in each of the compared test conditions, has been reported consistently before by others (Bronzini et al., 2012; Mercati et al., 2014; Garvican et al., 2014). Interestingly, Bronzini et al. (2012) assessed viability more closely during the first 24 h, i.e., at 3, 6, 9, 12, and 24 h (and further). They reported a steadily maintained viability up to 12 h; however, a steep decline in viability from 12 h transport time to 24 h was observed. It is also of no surprise that time (in our study 48 h vs. 72 h) appeared to have far less influence on viability of MSCs in a frozen state; this is in accordance with previous findings in equine MSCs (Garvican et al., 2014).

While there was no significant difference for temperature (4 °C vs. RT) in our study, an advantage of lower temperatures has been previously reported (Mercati et al., 2014). However, a higher rate of equine MSC survival at RT vs. 4 °C (and 37 °C) was reported by one research group before (Bronzini et al., 2012). There was also no influence of temperature in the negative temperature range (−20 vs. −80 °C) in our study. Thus, during transport, it is paramount to maintain temperatures at minimum below −20 °C. Temperatures of approximately −80 °C are preserved on dry ice but require additional effort and cost during shipment. A temperature of −20 °C may or may not be maintained with freezer packs and appropriate insulating packaging. Continuous temperature measurements within the transport box might be necessary for quality control as, to our knowledge, no equine studies are available concerning this practice. In clinical settings, transport of suspended equine MSCs at 4–10 °C was facilitated using freezer packs and insulated boxes (Godwin et al., 2012).

There are discrepancies in MSC viability from previous publications (Mercati et al., 2014; Garvican et al., 2014) compared to our findings. Interestingly, a markedly decreased viability after transport of suspended MSCs to less than 50% was observed by Bronzini et al. (2012) as well. Unfortunately, based on the given description of the automated counting, it is unclear whether the total cell count or just viable cells were determined. Further, differences between studies might be due to the MSC source (blood, adipose tissue, BM), the quantifying technique (manual vs. automated counting), individual settings within each laboratory, or other unknown factors. Direct comparison of protocols is impeded by curtailed description of protocols in some of the other studies (Bronzini et al., 2012).

The slightly decreased viability in our control group (time 0) was most likely due to logistic reasons: the samples for this group were handled together with the test groups, meaning that in part II, 29 samples were assessed at the same time. The resulting prolonged processing time may have influenced viability. This may partially account for the poor survival after freezing MSCs in part II of our study too, because one condition (CryoStor, −20 °C, 72 h), resulted in part II in 35.5% viability and in part III (where fewer samples were evaluated) in 61.8% viability. Because all samples within each part of the study were handled equally, comparison between individual settings in each part of the study are still valid in our opinion. This may indicate that, in the clinical setting, cells should be used as soon as possible after they have been thawed. Comparison with previous studies is not always possible, because control viability before starting the simulated transport was not specifically evaluated (Bronzini et al., 2012; Mitchell et al., 2015).

Also, considerable decrease of the total number of viable cells was observed in all test conditions in our study. It is important to distinguish between the influence of manipulation (placement into the syringe, subsequent withdrawal, cell counting procedure) alone and the actual loss in the transport container during transport. The control group of part III shows that just the manipulation accounts for a deficit of 28–43%; higher concentrated cell suspensions appeared to have less cell loss. We believe that the highest loss occurs during the cell counting procedure and therefore any conclusion regarding the cell numbers a patient receives directly from a transport container is not possible. A comparison to previous equine studies is not possible because similar data have not been published before.

The undisturbed adipogenic and osteogenic differentiation throughout the study is in accordance with previous findings (Mercati et al., 2014). We found that the chondrogenic differentiation appeared to be affected, though not significantly. Decreased chondrogenic differentiation may be of concern because MSCs are intraarticularly applied in horses for the treatment of osteoarthritis (Ferris et al., 2014) and further investigation into this possible effect is warranted.

Only negligible differences in recovered cell suspension volume were found when the five transport containers were compared. It has to be taken into account, that in part I the containers were tested at RT only; the outcome of the investigation might have been different at other temperatures, e.g., below freezing. The lack in differences between the containers was unexpected, because the material of our chosen transport containers varied considerably.

Counting the recovered cells after the simulated transport would have probably yielded further useful information. Cell count was determined in some (Bronzini et al., 2012; Mercati et al., 2014) but not in other (Garvican et al., 2014) previous studies evaluating the effect of transport on equine MSCs.

We noticed that the liquid content tended to foam after agitation, making aspiration in the cryotube and in the CellSeal more difficult. The foam formation in the glass syringes dissolved faster compared to all other containers. Agitation is necessary because cells tend to sediment quickly to the bottom of the container. We also noticed that even after agitation, cell accumulations sometimes remained visible in the fluid. This was irrespective of the type of container. Protein aggregation and particle formation in prefilled glass syringes is a known effect in the pharmaceutical industry (Jones, Kaufmann & Middaugh, 2005; Gerhardt et al., 2014). Prefilled glass syringes and some plastic syringes contain silicon oil as a lubricant to enable smooth plunger movement. It has been shown that the silicon oil-water and air-water interface combined with agitation are responsible for the aggregating effect (Jones, Kaufmann & Middaugh, 2005). Injecting large protein particles might be of concern in the equine patient, but has not been evaluated specifically. Also, risk of breakage of glass syringes compared to plastic containers has to be considered. In general, the use of syringes as transport containers seems appealing since they are ready-to-use products. However, they are not sterile on the outside and special precaution is required when using the syringes for sterile application. In contrast, a sterile syringe can be used to aspirate the content from cryotubes or CellSeal. We occasionally noticed that aspiration of small volumes from CellSeal was difficult and small pieces of plastic from the seal blocked the 21 G needle used for aspiration. A larger needle might have prevented that, but may result in injection of small plastic pieces into patients’ tissues. Additionally, the large port of cryotubes may be a potential entry for contamination during sample aspiration.

We compared various cell concentrations during transport, because injecting small volumes and therefore preparation of highly concentrated MSC suspensions might be of interest for the treatment of small tendon defects. In our experience (J Burke, 2015, unpublished observations) even 1 ml MSC suspension injected into small tendinous lesions (under sonographic guidance) leaked outside the defect and into the peritendinous tissue. This was proven by labeling the MSCs with SPIO (iron oxide particles) and subsequent sequential MRI evaluation. In clinical cases application of 2 ml volume at a concentration of 5 × 106 MSC/ml has been reported (Godwin et al., 2012; Smith et al., 2013). In previous experimental studies assessing MSC transport conditions comparatively lower concentrations of 0.5 × 106 MSC/ml (Bronzini et al., 2012), 1 × 106 MSC/ml (Mercati et al., 2014), 5 × 106 MSC/ml (Garvican et al., 2014), and 10 × 106 MSC/ml (Mitchell et al., 2015) were evaluated. In our study, no negative impact on MSC viability, proliferation and differentiation capacity was observed even in our highest concentrated solution of 20 × 106 MSC/ml.

The high number of comparisons within our study necessitated the application of Bonferroni correction. This fact combined with the observed high variability and low samples numbers are probably the reason for the lack of statistical significance.

Conclusion

In summary, we could not statistically prove any of our three hypotheses. However, transport media, transport time and the temperature during transport seem to be critical factors potentially influencing MSC quality.

Supplemental Information

Supplemental Information 1 Supplemental data part I

Click here for additional data file.

Supplemental Information 2 Supplemental data part II

Click here for additional data file.

Supplemental Information 3 Supplemental data part III

Click here for additional data file.

Additional Information and Declarations

Competing Interests

Author Contributions

Animal Ethics

Data Availability

The authors declare there are no competing interests.

Miguel Espina conceived and designed the experiments, performed the experiments, analyzed the data, contributed reagents/materials/analysis tools, wrote the paper, prepared figures and/or tables, reviewed drafts of the paper.

Henriette Jülke conceived and designed the experiments, analyzed the data, contributed reagents/materials/analysis tools, reviewed drafts of the paper.

Walter Brehm conceived and designed the experiments, contributed reagents/materials/analysis tools, reviewed drafts of the paper.

Iris Ribitsch conceived and designed the experiments, reviewed drafts of the paper.

Karsten Winter conceived and designed the experiments, analyzed the data, contributed reagents/materials/analysis tools, prepared figures and/or tables, reviewed drafts of the paper.

Uta Delling conceived and designed the experiments, analyzed the data, contributed reagents/materials/analysis tools, wrote the paper, prepared figures and/or tables, reviewed drafts of the paper.

The following information was supplied relating to ethical approvals (i.e., approving body and any reference numbers):

State Animal Care Committee: Landesdirektion Leipzig, Free State of Saxony, Germany; approval number V12/09.

The following information was supplied regarding data availability:

The raw data was supplied as Supplemental Table files.

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
