# Peer review of "Evaluation of transport conditions for autologous bone marrow-derived mesenchymal stromal cells for therapeutic application in horses"

_PeerJ, doi:10.7717/peerj.1773_

## Round 0.1 · original submission · Major Revisions

Please, amend the manuscript according to the suggestions of the three reviewers.

·

Basic reporting

The present study is a very interesting work that warrants publication after adressing some critical remarks:
1) How many replicates were used per circumstance?
2) The low viability after freezing is very unusual and in contrast with a recent publication from Texas A&M which reports more than 80% viability after freezing equine MSCs with both 5% and 10% of DMSO used (doi: 10.1186/s13287-015-0230-y). This warrants more explanation and raises some concerns about the freezing protocol (how did you freeze, was it temp monitored, were doses more than 5M/500ul too high for the used freezing system, were all carriers suitable for this, why were the cells frozen to -150°C first for part II of this study, etcetera). Please elaborate on and discuss these issues. In my opinion this warrants a more conservative interpretation of the results.
3) Include some pictures of the stainings per score and per staining.
4) More specific remarks are indicated in the uploaded pdf.

Experimental design

Good experimental design, yet more information is required.

Validity of the findings

The replications are important and a more conservative interpretation of the results because more explanation or replicates are needed to validate them.

Additional comments

It is a very interesting study and would benefit from some more explanation and a more conservative interpretation of the results.

Reviewer 2 ·

Basic reporting

The introduction to the field is not sufficient enough. The introduction should also present what is known of the clinical use of MSCs in humans, especially for graft-versus-host disease (GvHD)

Experimental design

The general question is relevant, but the design of the study is halting on several points. Major study weaknesses are:
-the omittance of cell number data in addition to viability to receive information about cell recovery
-unclear how MNCs were plated i.e. in which cell density after MNC isolation. Was this standardized at all?
-how was freezing really performed? What was the cell density at freezing? What was the temperature of the freezing media at freezing?
-how was cell thawing really performed? What was the temperature of the thawing media? How were the cells handled?
-what were the n-numbers in the experiments? And how many autologous MSC cultures were studied?

Validity of the findings

The data presentation should clearly indicate what was the n number in each experiment. Now it is very unclear and it is impossible to evaluate the significance and accuracy of the results. One major weakness in the conclusions is the failure to identify the parameters which are most crucial in cell freezing and thawing. The surprisingly low viablity of the frozen-thawed MSCs suggests that more focus should be put on the basic procedures in MSC freezing and thawing. Frozen MSCs usually exhibit a viability >80% after thawing if done properly.

Additional comments

The manuscript in its current form unfortunately needs major revision and new experimentation to gain validity. The low viability of the frozen MSCs suggests that the used protocols have not been optimized properly.

Reviewer 3 ·

Basic reporting

The paper “Evaluation of Transport conditions for autologous bone marrow-derived mesenchymal stromal cells for therapeutic applications in horses” is well writed with a good English language. The introduction and the background are clear and the literature are well referenced and relevant.The structure of the paper is conform to PeerJ standard and in the respect of PeerJ police.
The figure are relevant and of good quality and well described.

Experimental design

The research respect the scope of the journal and the paper is intersting . The investigation is performed with rigorous investigation and an high technical standard. Methods are well described and those can be clearly replicate.
The conclusion are well stated and linked with the current literature.

Validity of the findings

The benefit to literature is clearly stated. Data are robust, statistically sound and controlled.

Additional comments

The paper “Evaluation of Transport conditions for autologous bone marrow-derived mesenchymal stromal cells for therapeutic applications in horses” submitted to Peer J is very interesting work on an current topic. The study design is accurate and the manuscript is well written and presented . I am convinced that the study design allows to consider valid the results of this study and the authors were able to demonstrate the results of their study and they give also same guidelines for a better quality of the transport procedures of these biological materials used in horses and in other species in many countries in Europe and in USA.

---

## Round 0.2 · Minor Revisions

Please consider the final comment for the author from Reviewer 1.

·

Basic reporting

No comments (only for the discussion, cfr. general comments).

Experimental design

No comments.

Validity of the findings

No comments.

Additional comments

I would like to thank the authors for their thorough corrections and including very interesting data. The manuscript has hereby substantially improved. Especially the freezing protocol and histological images are highly appreciated and clarify a lot of issues.
Before final acceptance, I want to encourage the authors to omit the following lines: 471-475. Since the data are not significant, it is not worth spending more explanations and searching for correlations in the discussion. There are indeed studies claiming a toxic effect of DMSO on tendon and cartilage, but there are also many studies demonstrating no influence. I would therefore leave this issue and remove it from the discussion.
Furthermore, well done and congratulations with the manuscript!

---

## Round 0.3 · accepted · Accept

The paper has been improved and now can be published.